# The Immune System in Duchenne Muscular Dystrophy Pathogenesis

**DOI:** 10.3390/biomedicines9101447

**Published:** 2021-10-11

**Authors:** Luana Tripodi, Chiara Villa, Davide Molinaro, Yvan Torrente, Andrea Farini

**Affiliations:** Laboratorio di Cellule Staminali, Dipartimento di Fisiopatologia Medico-Chirurgica e dei Trapianti, Università degli Studi di Milano, Fondazione IRCCS Cà Granda Ospedale Maggiore Policlinico, Milano, Centro Dino Ferrari, Via Francesco Sforza 35, 20122 Milan, Italy; luana.tripodi@unimi.it (L.T.); chiara.villa2@unimi.it (C.V.); molinarodavide97@yahoo.com (D.M.); yvan.torrente@unimi.it (Y.T.)

**Keywords:** Duchenne Muscular Dystrophy, innate and adaptive immune system, inflammation

## Abstract

Growing evidence demonstrates the crosstalk between the immune system and the skeletal muscle in inflammatory muscle diseases and dystrophic conditions such as Duchenne Muscular Dystrophy (DMD), as well as during normal muscle regeneration. The rising of inflammation and the consequent activation of the immune system are hallmarks of DMD: several efforts identified the immune cells that invade skeletal muscle as CD4+ and CD8+ T cells, Tregs, macrophages, eosinophils and natural killer T cells. The severity of muscle injury and inflammation dictates the impairment of muscle regeneration and the successive replacement of myofibers with connective and adipose tissue. Since immune system activation was traditionally considered as a consequence of muscular wasting, we recently demonstrated a defect in central tolerance caused by thymus alteration and the presence of autoreactive T-lymphocytes in DMD. Although the study of innate and adaptive immune responses and their complex relationship in DMD attracted the interest of many researchers in the last years, the results are so far barely exhaustive and sometimes contradictory. In this review, we describe the most recent improvements in the knowledge of immune system involvement in DMD pathogenesis, leading to new opportunities from a clinical point-of-view.

## 1. Introduction

Muscular dystrophies (MDs) are a heterogeneous group of genetic diseases caused by mutations in proteins that mainly constitute the sarcolemma and the cytosol of the muscle fibers. Even if inflammation is a shared feature among MDs, differences exist in the molecular and cellular pathways involved in cellular infiltrates, suggesting inflammatory milieu differences in each form of MDs [1]. Duchenne Muscular Dystrophy (DMD) is a devastating X-linked disease caused by mutations in the dystrophin gene. The asynchronous cycles of muscle fibre degeneration in DMD exacerbate the muscle infiltration of macrophages and lymphocytes and their secretion of pro-inflammatory cytokines. The severity of muscle injury and the inflammation dictate the impairment of muscle regeneration and the successive replacement of myofibers with connective and adipose tissue [2]. No cure exists for DMD: corticosteroids are largely in routine use although limited by side effects [3]. In the last two decades, research efforts have been trying to identify small molecules able to bypass the wide range of dystrophin mutations, or cell populations with high in vivo homing capacity. The use of mini- and micro-dystrophin with recombinant AAV vectors showed promising results in terms of muscle force and the rescue of pathological phenotype, but it was limited by the worsening of clinical parameters and the activation of immune responses, namely the complement system [4]. Reducing the inflammatory features of muscular pathologies could represent a potential field able to bypass immune responses determined by AAV vectors [5].

In parallel, notable efforts were performed to identify muscle-resident and other non-muscular cells whose abilities were to be expanded *in vitro*; to migrate efficiently from the site of injection into dystrophic muscles; to develop into muscle and contribute to muscular regeneration. Different subpopulations were studied (muscle-derived stem cells; mesoangioblasts; CD133+ cells; mesenchymal stem cells) [6,7], but none of them were able to rescue efficaciously DMD phenotype, partly due to unresolved inflammatory responses that dramatically limited their migratory and differentiative potential [8]. 

### 1.1. Breaking the Immune Privilege in Skeletal Muscle 

During healthy muscle regeneration, there is a crosstalk between immune system and skeletal muscle, as demonstrated by the activity of macrophages that eliminate cellular debris, maintain muscle homeostasis and secrete factors regulating satellite cell (SC) proliferation [9,10]. In physiological conditions, these events are strictly and timely orchestrated to preserve the immune privilege that protects muscles from harmful inflammation (Figure 1A). In DMD, the abnormal regulation of regenerative processes leads to the rising of inflammation, the activation of the immune system and the consequent invasion of cytotoxic T-lymphocytes, neutrophils and macrophages that promote muscle damage [2,11] and cardiac dysfunctions [12,13,14] (Figure 1B).

### 1.2. Regulation of Inflammatory Process in DMD

Inflammatory mechanisms in DMD are commonly dependent on several modulators that in turn activate the aberrant proliferation of the immune cells. Due to loss of dystrophin, damaged myofibers release Danger Associated Molecular Patterns (DAMPs), while the increase of oxidative stress and defective calcium handling triggers the activation of innate immune cells as neutrophils and macrophages into skeletal muscles [2]. In particular, the over-expression of calcium in the cytoplasm determines the over-activation of calpain that induces NF-κB-dependent pro-inflammatory pathways. At the same time, the high concentration of ROS allows for the peroxidation of lipids in the sarcolemma of muscle fibers, enhancing membrane permeability [13]. 

These events determine the self-sustaining activation of the innate immune response. Constitutive MHC class I and II expressed on muscle cells activate professional antigen-presenting cells (APCs) that, in turn, present muscle antigens to T-CD4+ and T-CD8+ cytotoxic lymphocytes. This vicious circle is sustained by the presence of the inflammatory cells and the secretion of pro-inflammatory cytokines and chemokines that hinder the expression of anti-inflammatory Tregs and rapidly induce muscular necrosis [2] (Figure 1B). Thus, the balancing between innate and adaptive immunity in DMD pathophysiology was largely dissected in the last decade.

## 2. Unraveling Innate Immune Component in Muscular Dystrophies

### 2.1. Neutrophils 

The inflammatory event in mdx skeletal muscles is a timely process that starts the first two weeks of life and reaches a peak at 6–8 weeks. The dystrophic tissues are firstly invaded by Ly6C+/F4/80− neutrophils that (i) removes the debris originated from damaged myofibers; (ii) secretes TNF-α to improve the proliferation of Th1-cells; (iii) secretes myeloperoxidase (MPO) to activate the pro-inflammatory activity of macrophages; (iv) releases superoxide enhancing membrane lysis and granulocyte-differentiation antigen (Gr-1) [15]. The work of Kranig elegantly dissected the events promoting the recruitment of neutrophils in mdx muscle following well-regulated steps as rolling through the endothelium wall, the adhesion and finally the transmigration [16]. All these events in mdx mice are regulated by the increased expression of a plethora of proteins such as the receptor of advanced glycation end products (RAGE), intercellular adhesion molecule 1 (ICAM-1) and other chemokines as CD11α and β. In particular, the over-expression of the pro-inflammatory cytokine IL-6 and of the Lymphocyte function-associated antigen 1 (LFA-1) favor the recruitment of neutrophils into dystrophic muscles, allowing the development of inflammation and fibrosis. Accordingly, the use of drugs to deplete these cells was suggested as a feasible therapy to alleviate the dystrophic phenotype of DMD [17].

### 2.2. Innate Immunity and the Role of Macrophages 

During muscle repair, macrophages firstly acquire a pro-inflammatory phenotype (M1) and then an anti-inflammatory/regulatory profile (M2) that is necessary to maintain and solve the inflammatory process. In DMD, macrophages and neutrophils migrate into damaged muscles guided by pro-inflammatory signals constituted by chemokines and cytokines. Macrophages-derived NO causes the lysis of myofibers and worsens the sarcolemma ruptures, stimulating the release of free radicals from neutrophils [18]. In this scenario, macrophages are activated by specific inflammatory signals and trigger immune responses following the binding to Pathogen Recognition Receptors (as the Toll-like receptors) that are highly expressed on cytotoxic and regulatory T-cells [19]. In DMD, the complexity of the cellular and molecular interactions among the skeletal muscle cells and the immune system is determined by several steps, as the production of soluble moieties and the interactions of T-lymphocytes with other cells of the immune system. As an example, DMD muscle-resident macrophages abundantly express the matrix metalloprotease-9 (MMP-9) that allows for the over-expression of pro-inflammatory cytokines as IFN-γ, NF-κB and IL-6 [20]. In addition, M1-macrophages boost the secretion of TNF-α and the activation of necrotic pathways dependent on receptor-interacting protein kinase-1 (RIPK3) [21]. M1 macrophages inhibit IL-4 and block the proliferation of M2-macrophages: accordingly, the pro-myogenic hormone Klotho is drastically down-regulated, negatively mediating the activity of SCs [22]. Downstream targets of macrophages polarization include enhanced oxidative stress, TGF-β secretion and the expression of fibrotic genes [23]. 

In normal myofibers, fibroadipogenic precursor (FAP) cells interact with SCs secreting specific trophic factors that allow for tissue regeneration by supporting the differentiation of muscle stem cells. M1 macrophages eliminate the FAPs by rapid clearance to avoid the risk of excessive fibrosis deposition, and M2 macrophages stimulate the muscle commitment of SCs. In the DMD background, the over-expression of TGF-β determines FAPs survival proliferation and the differentiation into adipocytes and fibroblasts, expressing matrix genes and worsening the pathological phenotype [24]. 

In myofibers, Notch signaling pathways are fundamental for SCs activity, from quiescence to differentiation [25]. Notch can regulate negatively FAP’-adipogenesis through the expression of the Notch 2 receptor; the Notch-dependent expression of Delta1 and Jagged1 in myotubes can inhibit the adipogenic features of FAPs [26]. Interestingly, FAPs from young mdx mice do not suffer from Notch antiadipogenic effects, but they are not transformed into adipocytes according to the high concentration of TNFα that block their proliferation [27]. On the contrary, FAP differentiation is particularly critical in old mdx mice, where the decrease of pro-inflammatory cytokines quickly primes the adipogenic program, leading to massive fat deposition [26]. 

Natural Killer (NK) cells and monocytes are linked through a feed-forward amplification loop of T-bet/IFN-γ/IL-12 signalling, which causes the mutual activation of both NK cells and monocytes and fosters the recruitment of inflammatory cells to sites of inflammation [28]. Tregs coordinate M1 and M2 macrophage activation [29]: as described in detail in the 3.3. Chapter, Tregs produce high levels of IL-10 and presumably work to maintain tolerance with macrophage. In particular, they drive the polarization of M1 into M2 through TGF-β secretion [30]. Indeed, Tregs partly inhibit the expression of HLA-DR on macrophages, leading to down-regulation of proinflammatory cytokines in response to LPS stimulation [31]. 

### 2.3. Muscle Mast Cells: Exacerbation of Chronic Inflammation

In healthy muscle, mast cells are small quiescent cells located near the vessels of the endomysium. The proliferation and degranulation of mast cells are often accompanied by the release of pro-inflammatory molecules as histamine and TNF-α. Interestingly, mast cells were commonly found in the proximity of necrotic fibers and in ischemic areas [32,33]. 

Increased vascular permeability in DMD allows the muscular infiltration of immune cells; therefore, it has been suggested that mast cells are recruited from the circulation to muscle tissues where they enhance the ruptures of plasma membrane and exacerbate the pathological phenotype [34,35]. As expected, mdx mice treated with cromolyn to inhibit mast cell degranulation show myofiber strength [36] and decreased muscle necrosis [37].

### 2.4. Eosinophils and FAPs

Among the cells of the immune system, it was reported that eosinophils invade the muscle of DMD patients [38] and mdx mice [39] following increased cytotoxic T cells. In particular, it was reported that the number of these cells is strictly dependent on the expression of IL-5 cytokine: in dystrophic mice, eosinophils concentration increases at 4 weeks of age, remaining constant beyond 32 weeks even if the presence of T-lymphocytes is already down-regulated [39]. Vidal et al. suggested that excessive activity of eosinophils in the dystrophic context could provoke the abnormal proliferation of fibroblasts, worsening the pathological phenotype of DMD [40]. Recently, Sek et al. introduced a mutation on the dystrophin gene in PHIL mice lacking eosinophils, and they compared these mice with mdx mice and with mdx-IL5ko suffering from hypereosinophilia. Interestingly, they demonstrated that the eosinophils were not responsible for muscle damage and their expression was not correlated with the development of muscular impairment [41].

By investigating the dystrophic skeletal muscle lacking major basic protein-1 (MBP-1) [38] or eosinophil-deficient ΔdblGATA1 mice [42], the fundamental role of eosinophils in mediating Th1-Th2 transition and muscular regeneration through IL-4-mediated activation of FAPs was depicted [43]. When IL-4 is scarce, FAPs enhance fat deposition and differentiate into adipocytes, while in healthy condition IL-4 determines the transition of FAPs into fibroblasts and sustains myogenic differentiation [44]. 

Moreover, FAPs highly express interleukin 33 (IL-33), which recruits eosinophils in skeletal muscle, increasing muscle fibrosis through the secretion of group 2 innate lymphoid cells (ILC2s) and IL-5 [45]. 

In line with these results, glucocorticoids supplementation decreased the number of infiltrating eosinophils and, consequently, of IL-4 cytokine. This condition allows FAPs to differentiate into adipocytes and accumulate into injured muscles, leading to insulin resistance and muscle weakness [46]. 

### 2.5. Complement Activation

The complement system is a fundamental branch of the innate immune system, providing defense against pathogens through the activation of different biochemical pathways and the modulation of inflammatory cascades. It consists of three different pathways (classical, alternative, lectin-mediated) formed by twelve plasma proteins. Some of these proteins are up-regulated in mdx mice, increasing dysfunctions of the sarcolemma lacking dystrophin [47]. 

In the last years, different studies showed how the complement system functioned as a linker with the adaptive immune system, regulating effector and memory B-cells, CD4+ and CD8+ T-lymphocytes [48]. In the muscle of DMD patients—as in those of individuals suffering from muscular dystrophies and inflammatory myopathies—necrotic fibers showed the presence of protein agglomerates composed of membrane attack complexes and C5b-C9 complement components [49]. Accordingly, specific drugs that inhibit chemotactic complement factors could be useful to diminish the development of inflammatory cues or the number of necrotic fibers. Vila et al. showed that morpholino oligonucleotide-treated mdx mice expressed dystrophin, leading to the activation of an innate and adaptive immune system. In particular, they found anti-dystrophin antibodies that caused the mobilization of complement cascade and the over-expression of complement membrane attack complex on myofibers [50]. 

It is also likely that failures of myoblast transplantation trials were in part caused by the formation of membrane attack complexes that increased C3 convertase and induced the mobilization of macrophages and neutrophils, which led to cell transplanted death [51]. A strong complement activation is also engaged to the macrophages mediated immune response against AAV vectors [52], as observed in two different AAV trials with DMD patients (ClinicalTrials.gov: NCT03362502 and NCT03368742), which were therefore prematurely interrupted.

### 2.6. Cytokines and Chemokines

Inflammatory response is tightly regulated by adhesion molecules, signalling receptors, chemokines and cytokines—the family of interferons, transforming growth factors, TNF family of cytokines and interleukins—that are secreted by specific cellular populations and recruit immune cells from the blood stream into muscles. In DMD, chemokines and cytokines are responsible for abnormal innate and adaptive immune responses, muscle wasting, necrosis and fibrosis and for triggering the proliferation of pro-inflammatory cells as M1 macrophages, myo-fibroblasts and cytotoxic lymphocytes. Dystrophic patients and mdx mice exhibit dysfunctions in extracellular matrix (ECM) deposition and ECM-mediated cell interactions that guide an intense flow of inflammatory cells from the circulation into damaged muscle tissues. In DMD muscles, a high percentage of CD4+/CD8+ T-cells express CD49d isoforms that are commonly recognized as fibronectin receptors. Among DAMPs released from dystrophic muscle fibers, the High Mobility Group 1 Binding protein (HMGB1) protein binds to TLR4 on the surface of DCs and macrophages that turn into APCs. This way, APCs activate T lymphocytes through the MHC molecules-antigens bound, determining inflammation and muscle degeneration [53]. Similarly, the rising of oxidative stress in dystrophic muscles causes the up-regulation of NF-κB and its targets RAGE and the N-carboxymethyl lysine (CML). The modulation of targets by NF-κB activates in turn pro-inflammatory cytokines as interleukin-1β (IL-1β), interferon-γ (IFN-γ), IL-6, and other mediators as MAPKs, Jak/STAT and PI3K, leading to inflammatory responses [54]. Oxidative stress also induces the down regulation of hypoxia-inducible factor-1 (HIF-1) in DMD muscle, affecting the proper angiogenesis and the survival of endothelial cells [55]. 

Muscle endothelium highly expresses CCL14, CCL2, CXCL12 and CXCL14, which cause a severe recruitment of inflammatory cells as CD86+ and HLA-DR+ dendritic cells. Interestingly, it was demonstrated that muscle-derived Tregs in mdx mice up-regulate CCR1, whose ligands as CCL3, CCL5, CCL7 and CCL23 mediate the invasion of effector immune cells. Regenerating the muscular fibers of dystrophic muscles over-expresses the chemokine CXCL12, which in turn regulates the recruitment of cytotoxic lymphocytes and macrophages and the necrosis of myofibers [56].

Similar to NF-κB, the up-regulation of activator protein 1 (AP-1) is dependent on the pro-inflammatory cytokines and other growth factors, and its dysfunction affects cell death and proliferation [57].

### 2.7. Dendritic Cells 

Dendritic cells (DCs) are professional APC and represent a bridge between the innate and adaptive immunity. They were identified in non-lymphoid and lymphoid organs, where they mediate the formation of MHC-peptide complexes and cause antigen-specific responses of NKs and T- and B-lymphocytes [58]. Different works showed that injured muscles recruit macrophages, eosinophils and circulating DCs that remain in the tissues to promote muscle regeneration [59]. DCs regulate the commitment of T lymphocytes in Th1, Th2, Th17 and Tregs—and consequently their activity in a healthy and dystrophic muscle environment [60]. In addition, DCs recognize several self-molecules released from inflammatory cells by means of TLRs-mediated pathways. This way, the over-expression of pro-inflammatory cytokines and TLRs in DMD raised the question regarding the role of DCs in a dystrophic inflammatory background [61]. Even if their role was largely demonstrated in myositis, few studies proposed the participation of DCs in DMD pathogenesis through TLR7-mediated signals or the modulation of transforming growth factor (TGF)-β expression [58]. Other evidence demonstrated that the over-activation of CXCL14 in the muscular endothelium of DMD patients allows for the accumulation of activated CD86+ and HLA-DR+ DCs [62]. The combined expression of factor XIIIa and HLA-DRα identified DCs that regulate fibrotic development [63]. Moreover, DCs were down-regulated in DMD muscle biopsies following glucocorticoid therapy [64].

## 3. Adaptive Immune Response in DMD

Lymphocyte population does not play an important role in healthy muscle regeneration, due to the inability of skeletal muscle to activate a T cell response. However, it is likely that an inflamed muscle may constitute a milieu triggering HLA-DR and co-stimulatory molecule (for example CD40) upregulation and that myoblasts may work as optional APC cells, forming functional immunological synapses with T cells. Generally, T lymphocytes are found in degenerating muscles after acute trauma, and the recruitment into damaged muscle involves an adaptive immune response that normally depends on antigen exposure. Numerous observations have suggested that the presence of specific muscle autoantigens may drive the expansion of T lymphocytes and their activation [15]. Recent studies confirmed that immune system activation is partly dependent on lymphocyte activation and proliferation. The work of Vetrone et al. showed that antigen-specific T cells isolated from mdx muscles were able to undergo oligoclonal expansion [65]. Similarly, the detailed analysis of the T-cell receptor in the muscles of DMD patients [5] and mdx mice [66] revealed that immune cells (T-lymphocytes and Tregs) were able to recognize specific muscle-derived self-antigens. Mendell et al. demonstrated the unexpected presence of blood-derived auto-reactive CD4+ T lymphocytes against dystrophin epitopes in two patients before treatment [5]. Flanigan et al. showed that one third of 70 DMD patients’ cohort spontaneously developed a T-cell mediated immune response and that this event was significantly diminished following steroid treatment [67]. Accordingly, we proposed that few dystrophin-reactive T cells of DMD patients could escape from the thymus and be activated by dystrophin expressed from revertant myofibers in muscle tissues [68]. In addition, immunomodulation partly prevented the proliferation and priming of T-lymphocytes. Other works determined how the number of these autoreactive T-cells was dependent on both their unbalanced deletion in thymus or the activity of other inflammatory cells [69]. Since immunocompetent T-cells and Tregs are necessary for the maintenance of immune tolerance and mainly originate in the thymus [70,71,72], we demonstrated that the architecture of mdx thymus is severely impaired, according to the expression of ghrelin and autophagy machinery [68]. In particular, dysfunctions of the axis ghrelin (GHR)-GHR receptor (GHS-R) have been demonstrated to cause a reduction in the number of naïve T cells and consequent defects in thymic output [73]. These results suggest a potential involvement of GHS-R in the expression of genes associated with dystrophic thymic stromal environment changes and adipogenesis. Since GHR is also involved in T-cell maturation and the migration of T-cells from thymus [15,74], we showed that the transplantation of thymus derived from mdx mice into recipient nude mice determined the up-regulation of inflammatory and fibrotic markers and a marked metabolic breakdown with a consequent muscle atrophy and loss of force. Our results indicate that the involution of dystrophic thymus alters central immune tolerance and exacerbates muscular dystrophy. Specifically, transplanted dystrophic mdx thymus activated the host T lymphocytes of nude mice, resulting in increased fibrosis and reduced muscle performance, suggesting the importance of the adaptive immune system in DMD.

### 3.1. T-Cell Response in DMD Patients 

Onset inflammatory lesions of muscle tissues are mainly constituted of resident immune cells other than the cell populations that arrive at the site of damage during the activation of the inflammatory response. Heterogeneous cell infiltrates have been described not only in DMD patients but also in the pathogenesis of dystrophic animal models. In humans, immune cells are recruited in the muscles between the ages of 2 and 8 years old and are predominantly macrophages and T cells. B cell infiltration is usually rare [75]. In muscle injuries, CD4 + and CD8 + T cells play a very important role, as suggested by Steinman et al., who conducted studies on the muscle T cell population of DMD patients. They found that muscle T-cells from 12 DMD patients were characterized by a TCR (Vβ2 TCR), with antigenic specificity responsible for T cell selection and activation against a limited set of muscle antigens [76]. More interestingly, Mendell et al. demonstrated the unexpected presence of circulating CD4+ T lymphocytes against self-dystrophin epitopes in two patients before treatment [5]. Flanigan et al. showed that one third of a 70 DMD patient cohort spontaneously developed a T-cell mediated immune response against dystrophin, and this reaction was inversely dependent on steroid consumption [77].

### 3.2. T-Cell Response in DMD Animal Models 

In the muscles of mdx mice, the appearance of infiltrating immune cells was observed at 2–4 weeks of age, decreasing in severity by 3 months of age. One of the first studies was carried out by Spencer and co-workers, showing that the antibody-mediated inhibition of the CD8+ and CD4+ population promoted a decrease in muscle histopathology [78]. In line with this observation, we created a dystrophic mouse model lacking functional T and B lymphocytes (scid/mdx mouse) that developed a reduced diaphragmatic fibrosis at 1 year of age, accompanied by the downregulation of TGF𝛽 expression and an improvement in muscle regeneration [79]. Similarly, the absence of T-cells in *nu/nu*/mdx mice showed a fibrotic down-regulation in the diaphragm [80]. Vetrone et al. isolated from 1 month-old mdx muscle a V𝛽8.1/8.2 sub-population of T-lymphocytes that over-expressed the osteopontin that modulates the activity of neutrophils and Tregs, mediating the development of fibrosis [65]. The canine DMD model, the golden retriever muscular dystrophy (GRMD), suffers from a more severe phenotype than mdx mice, as pathological features are evident between 3 and 6 months of age, resembling those of DMD patients at 5–10 years [81]. In the last years, different works described biomarkers associated with the progression of the GRMD, as the over-expression of chitinase 3-like 1 (CHI3L1) [82] or the presence of CD49d+ circulating lymphocytes [83], useful to improve the clinical relevance of GRMD studies. In a preclinical context, we have intra-arterially transplanted autologous genetically corrected muscle derived CD133+ cells into GRMD dogs, which partially rescued the expression of dystrophin. However, this condition determined the emergence of a strong anti-dystrophin T-cell response [84]. Le Guiner and colleagues treated a cohort of 12 GRMD dogs with local and systemic injection of adenovirus specifically designed to express canine microdystrophin without pre-treatment immunosuppression. They found significant rescue of dystrophin expression and of pathological phenotype for over 2 years and, more interestingly, they did not describe any immune dysfunction in the GRMD dogs [85]. Lorant et al. showed that the injection of allogeneic muscular stem cells in GRMD dogs with partial immunosuppression significantly increased muscle force and repair, with nearly absent lymphocyte proliferation or anti-dystrophin immune responses [86]. 

### 3.3. Treg Cells

Since large numbers of Treg are needed to accomplish therapeutic efficacy, local injection at the site of inflammation (targeted delivery) may lower the numbers needed for therapy. Recently, it was discovered that intradermal injection of low dose Tregs inhibits skin inflammation into humanized mouse huPBL-SCID-huSkin allograft model [87]. Recent works from Gazzero et al. showed that the migration of Tregs into inflamed muscles could be dependent on ATP/P2X [88] and IL-33/ST-2 axis [89]. IL-33 is an alarmin produced by FAPs and skeletal muscle stem cells: its receptor is the interleukin-1 receptor-like 1 protein (also named ST2) that is expressed on CD4+ lymphocytes, Tregs, macrophages and FAPs. In addition, muscle-resident T regs secrete amphiregulin (AREG) that mediates the expansion of skeletal muscle stem cells in response to IL-33 [89]. Two origins have been described for FoxP3+ cells, whose numeric and functional importance remains in question. Most FoxP3+ cells differentiate in the thymus from immature Foxp3+CD4+ and CD8+ precursors (tTregs), as an alternative to conventional CD4+ T cells. The second (pTregs) occurs in the periphery, where the expression of FoxP3 in Tconv cells is primed by several triggers as the modification of histones and epigenetic signals. Since immunocompetent T-cells and Tregs are necessary for the maintenance of immune tolerance and mainly originate in the thymus, a fundamental question arose: are DMD-derived Tregs thymic-derived Tregs (tTregs) or do they differentiate in the periphery due to the inflammatory signals/triggers (pTregs) that are common in the DMD-inflammatory background? The identification of markers uniquely associated with tTregs or pTregs is still ongoing. The expression of the transcription factor Helios was thought to serve as marker of thymus-derived Tregs [90], but it is been recently known that it is expressed at substantial levels in pTregs induced peripherally by lymphopenia or agonist peptide and is generically induced upon the activation of CD4+ T cells. Similar contradictory results were obtained with other markers such as Neuropilin-1, LAP and LRRC32/GAMP [91]. TCR experiments clearly showed that tTregs predominantly represent the Tregs-pool in peripheral lymphoid organs. Similarly, Panduro et al. demonstrated that VAT Tregs and muscle Tregs have to be considered as tTregs, according to TCR expression [92].

tTregs express CD25, CTLA-4 and lymphocyte activation antigen-3 and are necessary to maintain self-tolerance so that their inhibition determines autoimmune dysfunctions. In the presence of TGF-β, naive-T cells can over-express the FoxP3 and achieve immunosuppressive features. The immunosuppressive functions of Tregs are driven by cell-to-cell contact mediated mechanisms or cytokine gradient as those mediated by TGF-β and IL-10 [93]. In accordance, following the expression of previously-cited surface markers, it was demonstrated that the binding receptor of CD25, the IL-2, mediates the proliferation of activated effector T cells [94], while the CTLA-4 blocks the activity of antigen-presenting cells [95]. Alternatively, it was shown that specific cytokines secreted by Th-subpopulations (as IFN-4 by Th2, STAT-3 by Th17 and T-bet by Th1) modulate the activity of Tregs, allowing immunosuppression. Interestingly, in the last years, it emerged that the function of Foxp3-expressing Tregs is not only limited to immunosuppression but is also dependent on their ability to convert into other Th-cell types according to specific cytokines and proteins expression. The effects of Tregs in muscle regeneration in mdx mice are driven by the modulation of proliferation and the functionality of muscle-resident Tregs and inflammatory cells. TCR can specificity contribute to the homing capacity of circulating Tregs into inflamed muscles, helping to attract and/or retain specific Tregs. Together with TCR expression, the homing and functionality of the Tregs subpopulation are determined by the specific range of chemokines/cytokines receptors. As an example, CCR10+ Tregs are recruited in cancer following CCL28 secretion, while Treg proliferation in gut lamina propria is dependent on the expression of CCR9+ and/or CX3CR1 [96]. According to the work of Panduro, three scenarios seem possible: (i) distinct tissue-Treg phenotypes are imprinted in the thymus during Foxp3+CD4+ T cell commitment; (ii) distinct tissue-Treg phenotypes appear after these cells are recruited into their home tissues following specific cues; (iii) a rough phenotype is imprinted in the thymus and it is specialized after Tregs homing [92].

## 4. Clinical Perspectives in DMD

Immunomodulatory therapies act through specific membrane markers that could be targeted. The inflammatory responses in DMD are driven by immune players secreted by specific cellular populations and are necessary to recruit immune cells from the blood stream into the muscular tissues (Figure 1B). Steroids possess an immunomodulatory capacity that allows for the rescue of muscle mass and strength, slowing down the appearance of symptoms and disease progression [97]. Unfortunately, important side effects such as weight gain, behavior issues, osteoporosis, high blood pressure and stomach dysfunctions are associated with steroids consumption. For this reason, their use in DMD patients is only permitted following medical prescriptions to optimize the beneficial effects of these drugs for a prolonged time [98,99] (Table 1). Since 2000, different therapeutic approaches emerged for several inflammatory and autoimmune diseases such as rheumatoid arthritis and multiple sclerosis, involving drugs that inhibit pro-inflammatory cytokines or modulate the proliferation of T-cells [2]. In DMD pathology, it is necessary to block the immune responses provoked by the inflammatory milieu and, at the same time, to preserve the reparative functions that are normally exerted by inflammation and eventually lead to myogenesis.

### 4.1. Blocking the Inflammatory Responses

As largely described, inflammatory events that characterize DMD patients are activated by dysfunctions in calcium channels and by the over-expression of ROS and NF-κB-dependent signaling factors, considered one of the most feasible therapeutic targets to alleviate inflammatory responses. To date, different drugs were tested to modulate the activity of proteins mediating the accumulation of calcium into the cells—such as the store-operated Ca2+ entry (SOCE), ryanodine receptors (RyRs) and the transient receptor potential (TRP) cation channel—or to diminish the gathering of antioxidants (Table 1). Similar interesting results were obtained with compounds blocking the NF-κB and other molecules as IKK (Table 1).

Rather than the inhibition of TNF-α, IL6 and IL1 that raised some concerns [78], interesting data were obtained with the blockade of cytokines and chemokines that mediate the innate immune responses, such as TLR7 and TLR9 and CXCL12/CXCR4 [100]. The anti-fibrotic effect of TGFβ neutralization was largely investigated (Table 1), but detailed and mechanistic approaches to understanding the pro-/anti-inflammatory pathways dependent on TGFβ are still missing [2]. As an example, the early inhibition of TGFβ diminished the amount of connective tissue in the diaphragm of mdx mice but, at the same time, up-regulated the pro-inflammatory CD4+ T cells [101]. Similarly, the depletion of TGFβ in the first weeks of dystrophic pathology blocked the proliferation of Tregs and worsened the pathological phenotype of muscle while defeating the development of fibrosis in older animal models [102]. Other anti-fibrotic compounds were evaluated, but definitive results were not obtained [103,104]. Disagreement on the correct age of the intervention, interplay among targeted pathways, cellular components linked on TGFβ expression and the unresolved investigation of epigenetic control on fibrosis development all render preliminary results hard to translate into humans.

Promising results were obtained through the inhibition of immunoproteasome subunits. We found that ONX-0194 treatment ameliorated the pathological phenotype of both the muscular and cardiac tissues of mdx mice [66,105]. 

Following the results of Mendell regarding the circulating dystrophin-reactive lymphocytes in DMD patients [5] or others coming from Tregs transplantation [106], it could be relevant to corroborate gene and cell therapies with the modulation of both Tregs proliferation and Tregs-dependent growth factors [107,108]. 

### 4.2. Immune Response against AAV Therapy 

AAV therapy for gene replacement or correction is scarcely applicable in DMD because of the size of dystrophin gene, the need of the systemic deliverance of AAVs to reach all the affected muscles of the body and the immunological drawbacks. In the last years, the amelioration of body-wide systemic gene transfer and the development of mini- and micro-dystrophin allowed for the more efficient widespread distribution of AAV-containing dystrophin through the entire body. However, severe limitations concerning the toxic responses at high doses of AAV vectors, the activation of the immune system and the definition of consistent patient cohorts (in terms of age, degree of pathology, dystrophin mutation) remained unsolved [109]. High-dose systemic AAV transplantation determines strong innate immunity responses that are weaker if a low-dose is intravascularly administered [110]. Similarly, adaptive immune reactions driven by CD4+ and CD8-T lymphocytes against the viral components were largely described in AAV-based clinical trials, but they were partially controlled by anti-inflammatory treatment with steroids [111]. Indeed, significant improvements must be accomplished to improve the safety and efficacy of this promising therapeutic approach.

### 4.3. Conclusions

Precise interventions on inflammatory pathways that affect the skeletal muscles of DMD patients prior to gene therapy are necessary to significantly augment the success of clinical interventions. Recently, it was demonstrated that after only six months following the injection of AAV-U7 to rescue dystrophin expression, the efficacy of this strategy dramatically worsened, probably due to dysfunctions of the dystrophic myofiber membranes. To improve dystrophic membrane integrity at the time of the injection of the adenoviruses, mdx mice were pre-treated with antisense oligonucleotides to allow for limited dystrophin expression at the sarcolemma [112]. These combined strategies—commonly referred to muscle priming—were used to improve the advantages of AAV therapy. In the same way, these approaches could be finalized at the elimination of the inflammatory cells that often destroy transplanted cells and at the down-regulation of pro-inflammatory mediators that partially reduce myofiber degeneration. To avoid the development of extensive muscle damage and the rising of inflammation and fibrosis, it is fundamental to treat patients as early as possible to increase the synergistic effects of different techniques leading to the improvement of dystrophin expression and, at the same time, to preservation of skeletal muscle from secondary dysfunctions due to dystrophin loss. Cell therapy represents a valuable theoretical alternative to drug treatment in order to replace defective adult muscle fibers and ameliorate pathological phenotype. Because of the difficulties in identifying a unique cellular candidate that retained high migration capacity, the efficient ability of homing and engraftment and the capacity of muscular differentiation, different subpopulations are now under investigation. An important clinical renovation could derive from the combination of gene and cell therapy, particularly the intra-arterial injections of patient autologous muscle stem cells treated to restore dystrophin production and to enhance the distribution of cells into the musculature.

Of potentially critical importance is the fact that factors modulating the type of inflammatory response might also play an essential role in determining the degree and severity of muscle degeneration. In line with this scenario, different works proposed that peripheral organs rather than skeletal muscle itself managed essential contributions in the development of inflammation. Accordingly, we described the involution of mdx thymus and found that this—when transplanted into recipient nude mice—led to dysfunctions of central immune tolerance, such as the up-regulation of inflammatory/fibrotic markers and marked metabolic breakdown determining muscle atrophy and loss of force. The implementation of the studies concerning the role of thymus in mediating dystrophic pathogenesis could lead to an understanding of whether DMD patients’ T cell development is impaired by specific dystrophin-mutations and whether defective T-cell proliferation could be rescued by specific genetic manipulation or the in vitro modulation of patient source cells that partly counteract thymic abnormalities. 

Indeed, it is defined that the development and normal homeostasis of both innate and adaptive immune systems is strictly associated to the gut microorganisms, the so-called microbiota. Interestingly, DMD patients suffer from several nutrition-related dysfunctions that impair body weight and composition, growth and energy requirements. Dietary metabolites function as immune-modulators: accordingly, nutritional interventions are the most effective and less expensive strategies and could represent a valuable co-adjuvant for DMD treatment by counter-acting the effects of chronic inflammation and oxidative stress.

## Figures and Tables

**Figure 1 biomedicines-09-01447-f001:**
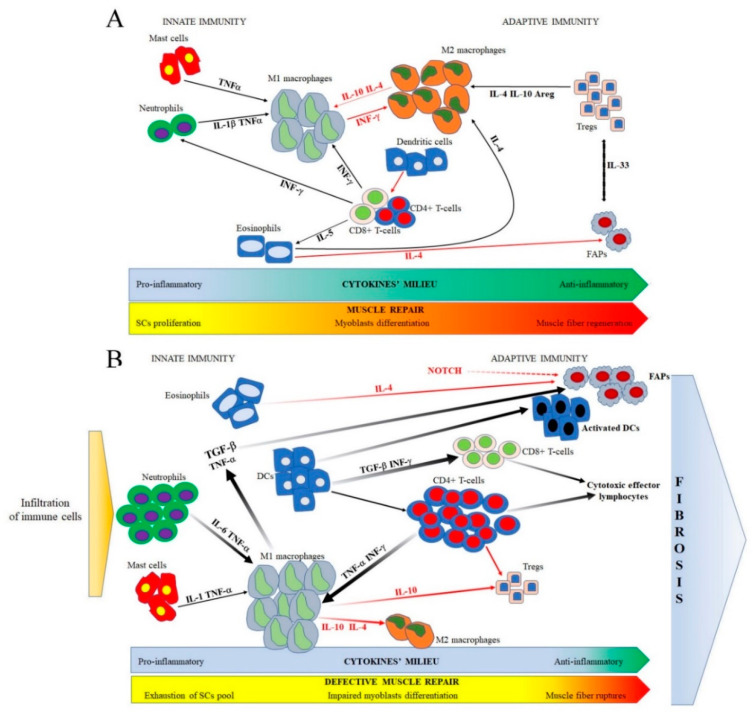
(**A**) During the initial phase of muscle repair in healthy individuals, the balanced expression of cells from innate immunity determines the expression of pro-inflammatory cytokines such as IL-6, TNFα and IL-1 that force muscle fibers necrosis and the clearing of cellular debris through M1 macrophages’ activation. In the later stage, the secretion of anti-inflammatory mediators such as IL-4, IL-10 and Areg promote the switching of macrophages phenotype to M2 and the proliferation of Tregs, while Tregs and FAPs are linked through the expression of IL-33. All these events lead to the resolution of inflammation and muscle regeneration. (**B**) In a dystrophic background, degenerating dystrophin-absent fibers secrete pro-inflammatory cytokines and chemokines: as a result, mast cells and neutrophils are recruited in the muscles where they activate the proliferation of M1 macrophages. At the same time, professional APCs as DCs activate the CD4+ CD8+ lymphocytes that, in turn, contribute to the release of TNFα, IL-6 and INF-γ and develop into cytotoxic T-cells. In this scenario, inflammatory cells determine the down-regulation of IL-10, IL-4 and Areg leading to the inhibition of anti-inflammatory cells such as M2 macrophages and Tregs. The over-expression of TGF-β mediated by M1 macrophages determines the activation of FAPs, whose proliferation is mediated by eosinophils according to the paucity of IL-4. All these cells contribute to the alteration of muscle homeostasis, muscle necrosis and the development of fibrosis. Specific cytokines with inhibitory effects are highlighted in red, while activating cytokines are highlighted in black; the thickness of arrows corresponds to the cytokines’ abundance in physiological and pathological muscle conditions.

**Table 1 biomedicines-09-01447-t001:** Immunomodulators of DMD pathology.

Drug & Target	Way of Action	Pathological Mechanism	Clinical Status
Remicade: NF-κB	Antibody against human TNF-α blocking NF-κB pathways	Inflammation	Preclinical
CAT-1004: NF-κB	NF-κB inhibitor	Inflammation	Phase I/II (^#^ NCT02439216)
VBP15: NF-κB	NF-κB inhibitor, increase sarcolemmal stability	Inflammation	Phase I (^#^ NCT02415439)
Etanercept: NF-κB	Inhibitor of soluble receptor to TNF-α	Inflammation	Preclinical
NEMO peptide: NF-κB	Inhibitor of NF-κB-dependent pathways	Inflammation	Preclinical
Flovocoxid: ROS	Antioxidant	Oxidative stress	Phase I terminated (^#^ NCT0133529)
Idebenone: ROS	Antioxidant	Oxidative stress	Phase III terminated, awaiting for commercial authorization (^#^ NCT01027884)
NAC	Antioxidant	Oxidative stress	Preclinical
Green tea extract	Antioxidant	Oxidative stress	Phase I (^#^ NCT01018615)
Tranilast: TRPV2	TRPV2 inhibitor	Calcium release	Authorized as anti-allergic drug
ARM210: RyR1	RyRs stabilizer, block calcium release from SR	Calcium release	Phase I
Infliximab: TNF-α	TNF-α inhibitor	Fibrosis	Preclinical
Losartan: angiotensin II	Inhibitor of Angiotensin II receptor antagonist	Fibrosis	Preclinical
Halofuginone: SMAD3	Inhibitor of the binding of SMAD3 with DNA	Fibrosis	Preclinical
Pirfenidone: TGF-β	Inhibitor of TGF-β signaling	Fibrosis	Preclinical
Deflazacort, Prednisone	Apoptosis, regulation of Ca^2+^ concentration and myogenesis, modulation of NF-κB	Inflammation	Phase III (^#^ NCT01603407)
Cyclosporine A: NFAT	Direct inhibition of T cell sub-populations and release of pro-inflammatory cytokines	Inflammation	Phase III (not effective)
Rapamycin	Reduction of infiltrating effector CD4+/CD8+ T cells in skeletal muscle tissue and activation of Tregs	Inflammation	Preclinical

^#^ unique identifier on ClinicalTrials.gov.

## Data Availability

Not applicable.

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
