# Peer review of "The Immune System in Duchenne Muscular Dystrophy Pathogenesis"

_biomedicines, 2021, doi:10.3390/biomedicines9101447_

Round 1

Reviewer 1 Report

The current work gives an extensive overview of the preclinical literature data in the domain of immune system in DMD. The paper is well structured and relates the actual knowledge between innate and adaptive immune response in DMD and concludes with the clinical perspectives in DMD. The review is composed from four distinct chapters and defines the main actors from the immune response in DMD.

Overall this is a very insightful review on a controversial subject; it could be interesting to bring some modifications in several chapters. For instance, the Introduction chapter could benefit the mention of available therapies and the large diversity of ongoing clinical therapies in DMD. Indeed, actual gene therapy replacement with AAV vector reported severe adverse events in several patients and it involved complement factor activation and thus immune system response interrogations in DMD. Moreover, the chapter 3 has an extensive bibliography and several citations from the author’s previous publication and looks more invested than the second chapter. Could the authors complete with additional literature data the chapter 2 and more in detail the subchapter Complement activation? In addition to some minor and major comment listed below, it would be interesting to add a subtitle in the Clinical perspectives in DMD chapter on AAV therapy immune response in DMD patients.

And last, for the Conclusion paragraph, the authors could hypothesize more extensively on the pathogenesis of immune response in the context of actual gene therapies with AAV vector based on the knowledge of their review work.

Major comments:

Line 53 chapter 1.2 lack reference

Line 353 which therapy precisely? Could the authors develop?

Line 361-362 could the authors reformulate in order to clarify their message.

Line 363 could the authors give the reasons why definitive results were not obtained in the author’s interpretation ?

Line 368-371 could the authors develop and give the objectives of this approach and also the difficulties of combined therapies?

Line 281 Could the authors give additional data in large animals as GRMD ?

Minor comments:

Line 34 and 47 reference

Line 348 please reformulate, as the steroids are recognized as standard of care in DMD and the therapy is prescribed on a life long manner taking into consideration the balance between the benefits and side effects adapted to the age of the patient.

Line 225 could the authors reformulate in order to clarify their message.

Author Response

Dear Reviewer, we appreciated your suggestions and accordingly we revised the Manuscript and we ameliorated each Chapter. 

Point-by-point response:

The current work gives an extensive overview of the preclinical literature data in the domain of immune system in DMD. The paper is well structured and relates the actual knowledge between innate and adaptive immune response in DMD and concludes with the clinical perspectives in DMD. The review is composed from four distinct chapters and defines the main actors from the immune response in DMD.

Overall this is a very insightful review on a controversial subject; it could be interesting to bring some modifications in several chapters. For instance, the Introduction chapter could benefit the mention of available therapies and the large diversity of ongoing clinical therapies in DMD. Indeed, actual gene therapy replacement with AAV vector reported severe adverse events in several patients and it involved complement factor activation and thus immune system response interrogations in DMD.

As suggested we modified the Introduction section describing new clinical challenges for DMD and – in particular – we focused on the development of gene therapy approaches for the disease

Moreover, the chapter 3 has an extensive bibliography and several citations from the author’s previous publication and looks more invested than the second chapter. Could the authors complete with additional literature data the chapter 2 and more in detail the subchapter Complement activation?

We add the new references in Chapter 2 and 3, and in particular we re-organized the Complement chapter

In addition to some minor and major comment listed below, it would be interesting to add a subtitle in the Clinical perspectives in DMD chapter on AAV therapy immune response in DMD patients.

As suggested, we add the subtitle in the new version of the manuscript

And last, for the Conclusion paragraph, the authors could hypothesize more extensively on the pathogenesis of immune response in the context of actual gene therapies with AAV vector based on the knowledge of their review work.

As suggested, we add the Conlcusion paragraph in the new version of the manuscript and we deeply discuss this point with other important considerations regarding the importance of secondary organs in the regulation of inflammation and immune responsed in DMD

Major comments:

Line 53 chapter 1.2 lack reference

We add the references as requested in the version of the Manuscript

Line 353 which therapy precisely? Could the authors develop?

This sentence was modified as suggested by the reviewer in the Chpater 4 in the new version of the Manuscript

Line 361-362 could the authors reformulate in order to clarify their message.

This sentence was modified as suggested by the reviewer in the Chpater 4 in the new version of the Manuscript

Line 363 could the authors give the reasons why definitive results were not obtained in the author’s interpretation?

As cited in the new version of the Manuscript, Disagreement on correct age of the intervention, interplay among targeted pathways, cellular components linked on TGF expression, and the unresolved investigation of epigenetic control on fibrosis development render preliminary results hard to translate into humans.

Line 368-371 could the authors develop and give the objectives of this approach and also the difficulties of combined therapies?

This part was modified in the new Chapter 4 in the new version of the Manuscript

Line 281 Could the authors give additional data in large animals as GRMD?

As requested, we modified the chapter of GRMD dogs and we described more in details the pre-clinical data obtained from studies involving these animals

Minor comments:

Line 34 and 47 reference

We add the references as requested in the version of the Manuscript

Line 348 please reformulate, as the steroids are recognized as standard of care in DMD and the therapy is prescribed on a life long manner taking into consideration the balance between the benefits and side effects adapted to the age of the patient.

We modified the sentence as follow in the version of the Manuscript:

“Steroids possess immunomodulatory capacity that allows rescue of muscle mass and strength, slowing down the appearance of symptoms and disease progression (Moxley et al., 2005). Unfortunately, important side effects as weight gain, behavior issues, osteoporosis, high blood pressure and stomach dysfunctions are associated to steroids consumption. For this reason, their use in DMD patients is only permitted following medical prescriptions to optimize the beneficial effects of these drugs for prolonged time (McDonald et al., 2017;Quattrocelli et al., 2017) (Table 1).”.

Line 225 could the authors reformulate in order to clarify their message.

As suggested we modified the sentence as follow:

Mendell et al demonstrated the unexpected presence of blood-derived auto-reactive CD4+ T lymphocytes against dystrophin epitopes in two patients before treatment (Mendell et al., 2010). Flanigan et al showed that one third of 70 DMD patients’ cohort developed spontaneously a T-cell mediated immune response and that this event was significantly diminished following steroid treatment (Flanigan, 2012). Accordingly, we proposed that few dystrophin-reactive T cells of DMD patients could escape from the thymus and be activated by dystrophin expressed from revertant myofibers in muscle tissues (Farini et al., 2021). In addition, immunomodulation partly prevented the proliferation and priming of T-lymphocytes

Reviewer 2 Report

With the success of gene therapy and the identification of driver genetic mutation and its consequence transcriptional defect of dystrophin, it would be interesting to mention how the well established pathway stimulate immune activation, barrier breakdown, and chronic immune dysregulation

Although immune involvement is less hyped about in the field of DMD and may be a critical field to be addressed. Authors need to point out the limitations of DMD gene as the only driver pathway, why gene therapy has not addressed current challenge or clinical discrepancies from a monogenic disease and compare the two fields' strength, and failures, to attract audience

May need more clinical real world data or clinical observations in the introduction before diving into the molecular basis of immune dysregulation and to summarize the authors point of view of whether immune dysregulation causes DMD, or immune dysreg is indirectly caused by downstream pathways of DMD.

What promises does targeting immune pathways in DMD bring? What are the future prospects? 

The review mentions stem cell in line 292, 295 and in the summary figures nevertheless fails to mention the implication in DMD. Considering that DMD is a germline mutation, what may be the benefitting effect of the stem cells in DMD patients carrying DMD faulty gene?

line 242, "Our results indicate that involution of dystrophic thymus exace bates muscular dystrophy by altering central immune tolerance. This represented the evidence that impaired immune system activation is a primitive feature of DMD independently from genetic muscle defects" is an overly assertive statement of linking central immune tolerance with DMD by solely the association of muscular dystrophy as DMD is only confirmed with genetic testing of DMD gene. 

Author Response

With the success of gene therapy and the identification of driver genetic mutation and its consequence transcriptional defect of dystrophin, it would be interesting to mention how the well established pathway stimulate immune activation, barrier breakdown, and chronic immune dysregulation

According to these suggestions, we discussed these points in the version of the Manuscript

Although immune involvement is less hyped about in the field of DMD and may be a critical field to be addressed. Authors need to point out the limitations of DMD gene as the only driver pathway, why gene therapy has not addressed current challenge or clinical discrepancies from a monogenic disease and compare the two fields' strength, and failures, to attract audience

We add several sentences regarding the gene theraphy in DMD in the new version of the Manuscript

May need more clinical real world data or clinical observations in the introduction before diving into the molecular basis of immune dysregulation and to summarize the authors point of view of whether immune dysregulation causes DMD, or immune dysreg is indirectly caused by downstream pathways of DMD.

We completely re-organized the Introduction section in the new version of the Manuscript

What promises does targeting immune pathways in DMD bring? What are the future prospects? 

We described the future perspectives in detail in the Chapter 4 in the new version of the Manuscript

The review mentions stem cell in line 292, 295 and in the summary figures nevertheless fails to mention the implication in DMD. Considering that DMD is a germline mutation, what may be the benefitting effect of the stem cells in DMD patients carrying DMD faulty gene?

We add a part in the new version of the Introduction section regarding the use of stem cell in DMD therapy

line 242, "Our results indicate that involution of dystrophic thymus exace bates muscular dystrophy by altering central immune tolerance. This represented the evidence that impaired immune system activation is a primitive feature of DMD independently from genetic muscle defects" is an overly assertive statement of linking central immune tolerance with DMD by solely the association of muscular dystrophy as DMD is only confirmed with genetic testing of DMD gene. 

We modified this sentence as follow:

“Our results indicate that involution of dystrophic thymus exacerbates muscular dystrophy by altering central immune tolerance. Specifically, transplanted dystrophic mdx thymus activated host T lymphocytes of nude mice resulting in increased fibrosis and reduced muscle performance, suggesting the importance of the adaptive immune system in DMD”

Reviewer 3 Report

In the present review Tripodi et al are offering an interesting compendium of information linked to the behavior of innate and adaptive immunity in DMD pathogenesis. Specifically, authors retrace the most relevant literature involving immune cells and dystrophies while summarizing recent findings that are poorly known to the new generation of myologists. The review is very well written and the different paragraphs, albeit digested quickly, are quite equilibrated in dealing on their respective theme. The comparison of the T-cell response between rodent dystrophic models and humans, has been particularly appreciated by this reviewer. However, some inaccuracies are present and should be solved prior to publication.

-A table summarizing immunomodulatory agents currently employed to mitigate DMD should be contemplated and opportunely added. Similarly, a table summarizing immunomodulatory agents under preclinical investigation should be included.

-line 107to109. The “new paradigm” should be clarified and properly described in order to highlight differences between canonical and the new macrophage population

-eosinophil paragraph. The reference Uezumi 2010 do not demonstrate that eosinophils are the immune population that activates FAPs. Such paper has been published by Heredia et al (2015).

IL4 released by eosinophils triggers FAP activation and phagocytic activities of these cells while preventing their fibro adipogenic differentiation.

Corticosteroid treatments seems to favor FAP adipogenesis by blunting eosinophils-derived IL4. Please add this information (Dong et al 2014).

-Information on the cross talk between immune cells and FAPs is not properly curated in this paper. In addition to the previous point, the manuscript lacks of the following information that are critical for the theme of the review.

  1. i) The balance between TNFa/TGFb, released by M1 and M2 macrophages, regulates FAP behavior during regeneration. Specifically, M1-derived TNFa induces FAP apoptosis at the end of the regeneration period by restoring FAP number to the initial concentrations. In mdx muscles the ratio TNFa/TGFb decreases and the increased concentrations of TGFb promote FAP survival as well as their differentiation into myofibroblasts (Lemos et al 2015).
  2. ii) dystrophic FAPs are also insensitive to notch ligand a condition that prompts these cells toward an adipogenic fate, thus aggravating dystrophinopathology. However, TNFa from leucocytes restores this sensitivity restricting their detrimental behavior in young dystrophic mice (Marinkovic2019 et al., Giuliani2021 et al)

-please modify the cell map in figure1 accordingly to these information

-please add in the cell map in fig1 an arrow linking FAPs and T-reg cells via IL-33

Author Response

In the present review Tripodi et al are offering an interesting compendium of information linked to the behavior of innate and adaptive immunity in DMD pathogenesis. Specifically, authors retrace the most relevant literature involving immune cells and dystrophies while summarizing recent findings that are poorly known to the new generation of myologists. The review is very well written and the different paragraphs, albeit digested quickly, are quite equilibrated in dealing on their respective theme. The comparison of the T-cell response between rodent dystrophic models and humans, has been particularly appreciated by this reviewer. However, some inaccuracies are present and should be solved prior to publication.

As suggested by this reviewer, we modified the Manuscript accordingly

-A table summarizing immunomodulatory agents currently employed to mitigate DMD should be contemplated and opportunely added. Similarly, a table summarizing immunomodulatory agents under preclinical investigation should be included.

We add a comprehensive Table 1 in the new version of the Manuscript indicating the immunomodulatory agents currenlty employed in the DMD. Moreover, we specified the preclinical/clinical status of the studies in which these drugs are involved.

-line 107to109. The “new paradigm” should be clarified and properly described in order to highlight differences between canonical and the new macrophage population

As suggested, we modified this part in the new version of the Manuscript

-eosinophil paragraph. The reference Uezumi 2010 do not demonstrate that eosinophils are the immune population that activates FAPs. Such paper has been published by Heredia et al (2015).

As suggested, we modified this reference in the new version of the Manuscript

IL4 released by eosinophils triggers FAP activation and phagocytic activities of these cells while preventing their fibro-adipogenic differentiation.

As suggested, we expanded the FAPs chapter in the new version of the Manuscript

Corticosteroid treatments seems to favor FAP adipogenesis by blunting eosinophils-derived IL4. Please add this information (Dong et al 2014).

As suggested, we modified this sentence in the new version of the Manuscript

-Information on the cross talk between immune cells and FAPs is not properly curated in this paper. In addition to the previous point, the manuscript lacks of the following information that are critical for the theme of the review.

  1. i) The balance between TNFa/TGFb, released by M1 and M2 macrophages, regulates FAP behavior during regeneration. Specifically, M1-derived TNFa induces FAP apoptosis at the end of the regeneration period by restoring FAP number to the initial concentrations. In mdx muscles the ratio TNFa/TGFb decreases and the increased concentrations of TGFb promote FAP survival as well as their differentiation into myofibroblasts (Lemos et al 2015).
  2. ii) dystrophic FAPs are also insensitive to notch ligand a condition that prompts these cells toward an adipogenic fate, thus aggravating dystrophinopathology. However, TNFa from leucocytes restores this sensitivity restricting their detrimental behavior in young dystrophic mice (Marinkovic2019 et al., Giuliani2021 et al)

We add extensive FAPs description in Eosinophils paragraph (2.4) and in the chapter dedicated to macrophages

-please modify the cell map in figure1 accordingly to these information

-please add in the cell map in fig1 an arrow linking FAPs and T-reg cells via IL-33

According to suggestions of the referee, we modified the Figure 1A and 1B

Round 2

Reviewer 2 Report

Very detailed feedback

Modified manuscript to key areas of concern